# A symbiotic bacterium of shipworms produces a compound with broad spectrum anti-apicomplexan activity

Roberta M. O'Connor[1]*, Felix J. Nepveux V[2], Jaypee Abenoja[1], Gregory Bowden[1], Patricia Reis[1], Josiah Beaushaw[1], Rachel M. Bone Relat[1], Iwona Driskell[1], Fernanda Gimenez[1], Michael W. Riggs[3], Deborah A. Schaefer[3], Eric W. Schmidt[4], Zhenjian Lin[4], Daniel L. Distel[5], Jon Clardy[6], Timothy R. Ramadhar[7], David R. Allred[8], Heather M. Fritz[9], Pradipsinh Rathod[10], Laura Chery[10], John White[10]

1 Department of Veterinary Microbiology and Pathology, College of Veterinary Medicine, Washington State University, Pullman, Washington, United States of America, 2 Division of Geographic Medicine and Infectious Diseases, Tufts Medical Center, Boston, Massachusetts, United States of America, 3 School of Animal and Comparative Biomedical Sciences, University of Arizona, Tucson, Arizona, United States of America, 4 Department of Medicinal Chemistry, University of Utah, Salt Lake City, Utah, United States of America, 5 Ocean Genome Legacy Center, Northeastern University, Nahant, Massachusetts, United States of America, 6 Department of Biological Chemistry and Molecular Pharmacology, Harvard Medical School, Cambridge, Massachusetts, United States of America, 7 Department of Chemistry, Howard University, Washington DC, United States of America, 8 Department of Infectious Diseases and Immunology, College of Veterinary Medicine, and Emerging Pathogens Institute, University of Florida, Gainesville, Florida, United States of America, 9 California Animal Health and Food Safety Lab, University of California, Davis, California, United States of America, 10 Department of Chemistry, University of Washington, Seattle, Washington, United States of America

* rob.oconnor@wsu.edu

**Data Availability Statement:** All relevant data are within the manuscript and its Supporting Information files.

## Abstract

Apicomplexan parasites cause severe disease in both humans and their domesticated animals. Since these parasites readily develop drug resistance, development of new, effective drugs to treat infection caused by these parasites is an ongoing challenge for the medical and veterinary communities. We hypothesized that invertebrate-bacterial symbioses might be a rich source of anti-apicomplexan compounds because invertebrates are susceptible to infections with gregarines, parasites that are ancestral to all apicomplexans. We chose to explore the therapeutic potential of shipworm symbiotic bacteria as they are bona fide symbionts, are easily grown in axenic culture and have genomes rich in secondary metabolite loci [1,2]. Two strains of the shipworm symbiotic bacterium, *Teredinibacter turnerae*, were screened for activity against *Toxoplasma gondii* and one strain, T7901, exhibited activity against intracellular stages of the parasite. Bioassay-guided fractionation identified tartrolon E (trtE) as the source of the activity. TrtE has an $EC_{50}$ of 3 nM against *T. gondii*, acts directly on the parasite itself and kills the parasites after two hours of treatment. TrtE exhibits nanomolar to picomolar level activity against *Cryptosporidium*, *Plasmodium*, *Babesia*, *Theileria*, and *Sarcocystis*; parasites representing all branches of the apicomplexan phylogenetic tree. The compound also proved effective against *Cryptosporidium parvum* infection in neonatal mice, indicating that trtE may be a potential lead compound for preclinical development. Identification of a promising new

**Funding:** This study was funded by National Institutes of Health, National Center for Complimentary and Integrated Health (grant numer: R21AT009174; https://www.niaid.nih.gov/) to RMO. Research reported in this publication was supported by the Fogarty International Center of the National Institutes of Health under Award Number U19TW008163 to EWS. The content is solely the responsibility of the authors and does not necessarily represent the official views of the National Institutes of Health (https://www.fic.nih.gov/Pages/Default.aspx). The funders had no role in study design, data collection and analysis, decision to publish, or preparation of the manuscript.

**Competing interests:** The authors have declared that no competing interests exist.

compound after such limited screening strongly encourages further mining of invertebrate symbionts for new anti-parasitic therapeutics.

## Author summary

Apicomplexans are intracellular protozoan parasites that cause significant disease in humans and the livestock we rely on for food. Because these parasites easily develop drug resistance, new drugs are always needed. To identify new anti-apicomplexan drugs, we investigated the compounds produced by symbiotic bacteria of shipworms, marine mollusks that burrow into and eat wood. We screened shipworm symbiotic bacteria for anti-parasitic activity and identified a compound, tartrolon E, with potent, rapid, highly selective and irreversible activity against parasites representing all branches of the apicomplexan tree. This compound was also highly effective in neonatal mice against infection with the intestinal apicomplexan parasite, *Cryptosporidium*. This study describes the first pan-anti-apicomplexan compound and unveils an unexplored source of anti-parasitic compounds.

## Introduction

Apicomplexan parasites cause devastating disease in humans and their domesticated animals. Control and treatment of these parasitic infections has proven challenging for multiple reasons. *Plasmodium* species, the causative agents of human malaria, are masters of immune system evasion [3] and rapid development of drug resistance [4]. *Toxoplasma gondii* infects all mammalian species, and establishes a latent infection that remains untreatable [5]; active infection with this parasite during pregnancy can cause severe birth defects [6]. *Cryptosporidium sp. are* among the most common causes of childhood diarrhea worldwide [7], and yet the current drug arsenal consists of one poorly efficacious drug [8]. Apicomplexans that affect domesticated animals, such as *Babesia*, *Theileria*, and *Eimeria* species are developing resistance to the drugs used to control them [9–11], with few to no new drugs in the pipeline [12].

For centuries, nature has provided an abundance of effective therapeutics. In the past 30 years, two-thirds of antimicrobial drugs have been derived or conceived from natural products, even with the advent of synthetic chemical libraries and targeted drug development [13]. The marine environment is the next great frontier in natural product drug discovery [14]. Many of the marine compounds discovered thus far are produced by the symbiotic bacteria of marine animals [15]. One highly unusual marine symbiosis is that of the shipworm, a marine mollusk that burrows into and eats wood [16]. Unlike other wood-eating animals, the shipworm gut lacks a microbial community [17]. Instead, intracellular bacteria within cells of the gill produce cellulolytic enzymes that are exported out of the host cell, out of the gill and to the lumen of the gut to digest the shipworm's woody diet [2]. The genomes of these shipworm symbionts are also rich in secondary metabolite loci [1] which are expressed within the shipworm [18,19] and produce compounds that likely play a role in maintaining the sterility of the gut and protecting the animal from pathogens.

Since secondary metabolites produced by shipworm symbionts must move through multiple cellular compartments and between organs to support the symbiosis, we reasoned that these symbionts could be a rich source of therapeutics effective against intracellular pathogens. Furthermore, there is a reasonable ecological rationale for mining mollusk symbiont

compounds for activity against apicomplexan parasites Mollusks are susceptible to infection with gregarines [20,21], apicomplexan parasites that are ancestral to all other apicomplexans [22]. While many gregarines are non-pathogenic, some are known pathogens for their invertebrate hosts [23]. Mollusk symbionts may well have evolved strategies to help protect their hosts against invasion by pathogenic gregarines. Thus, we reasoned that the secondary metabolite repertoire of shipworm symbionts could include compounds effective against apicomplexan parasites and potentially compounds that target ancestral processes common to all apicomplexans. Another important aspect of the shipworm symbiosis, critical for testing this hypothesis, is that bacteria demonstrated to be bona fide members of the shipworm gill symbiont community have been identified, characterized and brought into axenic culture [1,24].

In this study we screened two strains of the shipworm gill symbiont, *Teredinibacter turnerae*, and identified a compound with potent activity against parasites representing all branches of the apicomplexan phylogenetic tree. These results support our hypothesis and highlight a new approach for anti-parasitic drug discovery.

## Results

### Culture medium, in which the shipworm symbiont *Teredinibacter turnerae* T7901 was grown, kills intracellular *Toxoplasma gondii*

For our initial study, we chose to look for anti-apicomplexan compounds in *Teredinibacter turnerae* [25]. *T. turnerae* is a gill symbiotic bacterium common to many species of shipworms [24]. We grew two strains of *T. turnerae*, T7901 [1] and T7902 [24], in shipworm bacterial media (SBM), pelleted the cells and tested the culture supernatants for activity against *T. gondii.* We chose *T. gondii*, a ubiquitous coccidian parasite, as a target parasite because it represents a significant human and animal pathogen, it is easily propagated in vitro and is amenable to genetic manipulation. *T. gondii* is often used as a model for other, more challenging apicomplexans [26]. *T. gondii* RH strain tachyzoites were allowed to establish infection in human foreskin fibroblasts (HFF) for 24 hours, then exposed to T7901 or T7902 culture supernatants by addition of the supernatants to the parasite cultures at final dilutions of 1:100, 1:250 or 1:500. Parasites were evaluated after 24 hours of treatment by immunofluorescence (IFA) microscopy with rabbit antibody to the *T. gondii* surface antigen, SAG1 [27]. Culture supernatant from T7902 had no effect on parasites at any dilution tested, but culture supernatant from T7901 inhibited growth of intracellular stages of *T. gondii* even at a 1:500 dilution (Figs 1 and S1). No morphologically intact parasites could be observed within the parasitophorous vacuoles, and SAG1, while detectable within the vacuoles, was amorphously distributed (Figs 1 and S1).

### Tartrolon E is the source of anti-*T. gondii* activity

To identify the source of the anti-parasitic activity, bioassay-guided fractionation of *T. turnerae* T7901 culture supernatant was initiated. Ethyl acetate extracts of culture supernatant were dried, dissolved in methanol and eluted in seven fractions from a C18 column, using a methanol/water step gradient. The fractions were tested for activity against *T. gondii* using a monolayer protection assay. In this assay, parasite growth is monitored by evaluating the condition of the host cell monolayer. When the parasites replicate they destroy the host cell monolayer, but if parasite replication is inhibited, the host cell monolayer remains intact. Anti-*T. gondii* activity was primarily observed in the fractions eluted with 80–90% methanol (S2 Fig). Since these fractions were known to be rich in tartrolon E (trtE), an anti-bacterial compound previously isolated from *T. turnerae* T7901 [18] (structure shown in Fig 2A), trtE was purified (S3–S6 Figs) as previously described from bacterial pellets (purification method 1) [18].

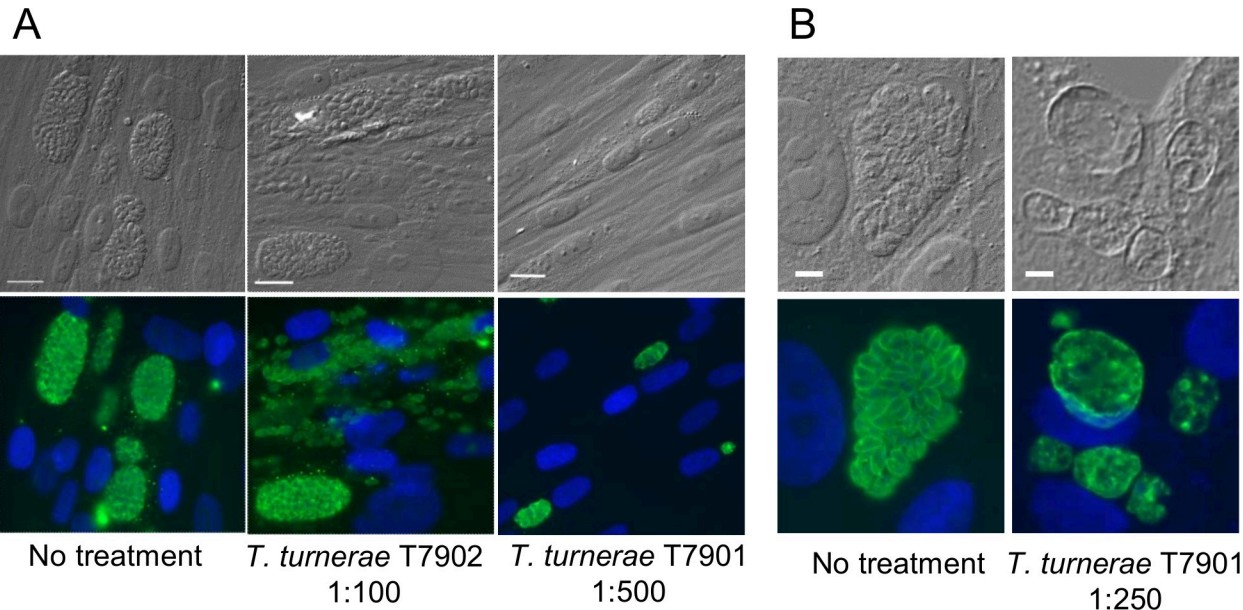

**Fig 1. *Teredinbacter turnerae* T7901 produces a compound with activity against intracellular *Toxoplasma gondii*.** *T. turnerae* strains T7901 and T7902 were grown in SBM for 9 days. Supernatants from the bacterial cultures was added to HFF cells infected with *T. gondii* at 1:100, 1:250 or 1:500 dilutions in cell culture medium and IFAs performed 24 hours later. Upper panels: DIC images, Lower panels: green, rabbit anti-SAG1 antibody, blue: DAPI. Additional representative images are shown in S1 Fig. **A.** T7901, but not T7902, supernatants inhibited the growth of *T. gondii*. Scale bar = 16 μm applies to both upper and lower images. **B. Left panels**: In DMSO treated samples individual tachyzoites are visible within the parasitophorus vacuole. **Right panels**: In trtE treated samples, the parasite vacuoles remained intact, but individual intact tachyzoites were no longer identifiable in the parasitophorous vacuole. Scale bar = 5 μm applies to both upper and lower images.

Purified trtE inhibited intracellular *T. gondii* growth, and similar morphological aberrations were observed in parasites treated with either T7901 culture supernatant (Figs 1 and S1) or purified trtE (Figs 2B and S7).

## TrtE is highly potent against intracellular *T. gondii*

To obtain sufficient trtE for in vitro and in vivo experiments, bacterial growth conditions were optimized, and a modified purification method was developed and validated by ESI-HRMS, LC/MS/MS and NMR (purification method 2; see methods and S8–S9 Figs). To determine the effective concentration at which trtE inhibited the growth of 50% of *T. gondii* parasites ($EC_{50}$), HFF cells were seeded into 96 well plates and infected with luciferase expressing *T. gondii* tachyzoites. 24 hours post-infection two-fold serial dilutions of trtE, purified by method 1 or method 2, or DMSO vehicle control were added to the infected cells. 24 hours later, the infected cells were lysed and parasite growth quantified by luciferase expression. The $EC_{50}$ of trtE against intracellular *T. gondii* was determined to be 3 nM regardless of purification method. The compound was equally effective against two strains of *T. gondii*, ME49, a slow growing, clinically relevant strain, and RH, a rapidly growing laboratory adapted strain (Fig 2C).

## TrtE is effective against extracellular *T. gondii* tachyzoites

Because we had tested the compound against intracellular parasites, it was unclear if trtE was exerting a direct effect on the parasites, or if trtE acted indirectly on the parasites by modulating host cell pathways. To determine if trtE had a direct effect on the parasite itself, extracellular *T. gondii* tachyzoites were incubated for varying times and with varying concentrations of trtE or DMSO vehicle control, then the compound was washed away and the parasites added

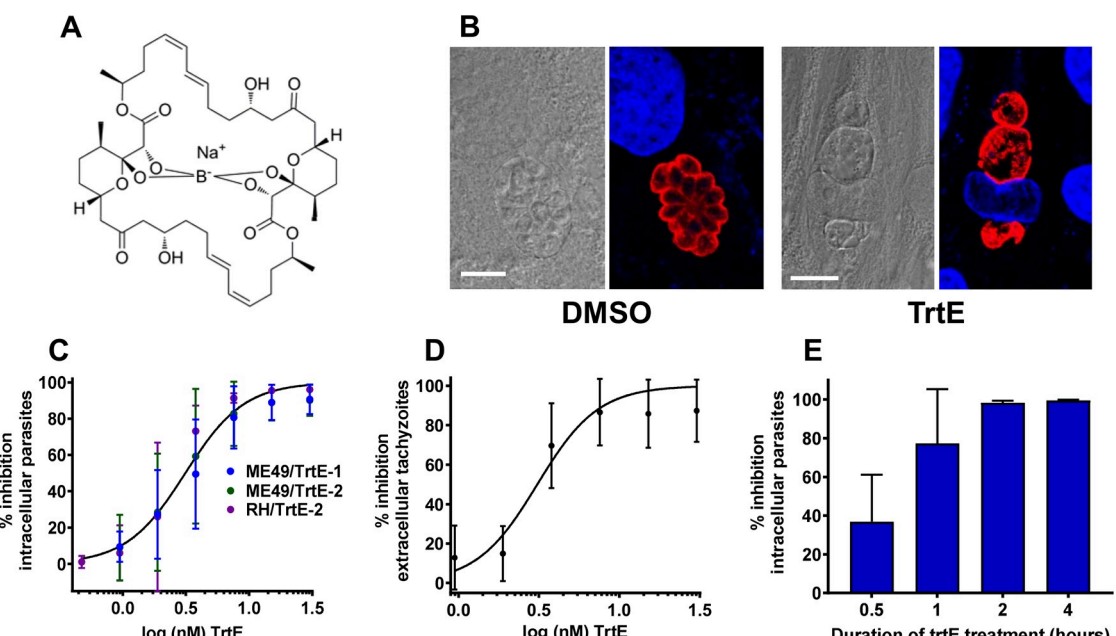

**Fig 2. Tartrolon E has potent and rapid parasiticidal activity against intracellular and extracellular *Toxoplasma gondii*: A.**
Structure of trtE. **B.** HFF cells infected with *T. gondii* RH strain parasites were treated with DMSO vehicle control (left panels) or 60 nM trtE (right panels) for 24 hours. Infected cells were then fixed and processed for IFA. Left images show DIC images, right images show corresponding IFA (red = rabbit anti-*T. gondii* SAG1 antibody, blue = DAPI). Scale = 5 μm. Images are enlarged from S7 Fig, panels A and B. The morphology of trtE-treated parasites is similar to that seen with T7901 culture supernatant (compare Figs 1 and 2, and S1 and S7). **C.** Luciferase expressing RH or ME49 strain *T. gondii* parasites were allowed to infect HFF cells for 24 hours, then the infected cells were treated with two-fold dilutions of trtE purified by method 1 (TrtE-1) or method 2 (TrtE-2). Parasite growth was quantified 24 post-treatment by luciferase expression. Data were analyzed using the log[inhibitor]vs response-Variable slope (four parameter) regression equation in Graphpad Prism and curves compared using the extra sum of squares F-test. One curve adequately fit all the data sets (p = .42) with an estimated $EC_{50}$ of 3.0 nM (95% CI of 2.5 to 3.5). **D.** Luciferase expressing RH strain tachyzoites, isolated from their host cells, were treated with two fold dilutions of trtE or DMSO control for 2 hours, then the compound was washed off and the parasites allowed to infect HFF cells for 24 hours. Parasite growth, as evaluated by luciferase expression, was inhibited by trtE treatment of tachyzoites with an $EC_{50}$ of 3.1 nM (analysis as for **C**; 95% CI of 2.3–4.3). **E:** HFF cells were infected with luciferase expressing *T. gondii* ME49 strain for 24 hours. Infected cells were then treated with 30 nM trtE or DMSO control for 0.5 to 4 hours, at which times the compound was washed off the infected cells. Parasite growth was evaluated by luciferase expression 72 hours after treatment. Parasites were killed after 2 hours of trtE treatment.

to host cells. Growth of the parasites was measured 24 hours later by luciferase expression. Extracellular tachyzoites treated with trtE for 2 hours were unable to establish infection, with an $EC_{50}$ of 3.1 nM (Fig 2D), which was not significantly different from the $EC_{50}$ of trtE against intracellular stages, consistent with the compound acting directly on the parasites.

## TrtE rapidly kills *T. gondii*

*T. gondii* parasites treated with trtE exhibited rapid inhibition of parasite growth, coalescence of the parasites within parasitophorous vacuole, and apparent expansion and fusion of parasitophorous vacuoles within cells (S1 Movie; compare to DMSO treated parasites over the same time period, S2 Movie). Since trtE rapidly caused disintegration of tachyzoites, we hypothesized that the compound was parasitcidal. To test this hypothesis, we infected HFF cells with luciferase-expressing *T. gondii* ME49 for 24 hours, then treated infected cells with trtE for 30 minutes to 4 hours. The compound was then washed off the infected cells, and parasite growth was quantified 3 days after the removal of the compound. Parasites treated for at least 2 hours were killed by trtE (Fig 2E).

## TrtE exhibits broad spectrum anti-apicomplexan activity in vitro

We had postulated that secondary metabolites generated by symbiotic bacteria of invertebrates might include compounds that target gregarines, an ancient group of apicomplexans found in most, and perhaps all, invertebrates [23]. The only gregarines known to infect mammals are *Cryptosporidium* species [28], gastrointestinal parasites that cause diarrheal disease in humans and other mammals [29]. HCT8 cells were infected with luciferase-expressing *Cryptosporidium parvum* parasites [30] for 24 hours, then two-fold dilutions of trtE added to the infected cells. 48 hours after the addition of trtE, parasite growth was quantified by luciferase expression [30,31]. Intracellular replication of *C. parvum* was inhibited by nanomolar concentrations of trtE (Fig 3A, $EC_{50}$ = 3.85 nM;). To examine the morphology of trtE treated *C. parvum* parasites, HCT8 cells were infected with *C. parvum* oocysts for 8 hours. Extracellular parasites were washed off and medium containing 60 nM trtE added to the infected cells. After 12 hours of treatment, infected cells were fixed and processed for IFAs. IFAs showed that DMSO-treated parasites formed normal type I meronts (Fig 3B upper panels, and S10 Fig, panels A, B and C). In trtE treated cultures the few parasitophorous vacuoles observed contained the *C. parvum* surface antigen gp15 but no visible intact merozoites (Fig 3B lower panels and S10 Fig, panels D, E and F).

Since trtE was highly effective against *C. parvum*, we tested the compound for activity against five other apicomplexan parasites representing the piroplasms (*Babesia bovis*, *B. bigemina* and *Theileria equi*), the hemoparasites (*Plasmodium falciparum*), and *Sarcocystis neurona*. Two fold dilutions of trtE were tested against these parasites in in vitro drug response assays conducted as previously described [30,32–38] or as described in the methods. All five apicomplexans tested were susceptible to trtE. $EC_{50}$ values were calculated from the inhibition curves shown in Fig 3C (*Babesia* sp), Fig 3E (*Plasmodium falciparum*) and Supplemental Fig S11 Fig (*Sarcocystis neurona* and *Theileria equi*) and ranged from 15.9 nM (*B. bovis*, Fig 3C) to 105 pM (*P. falciparum*, Fig 3D).

To evaluate the morphological changes in one of the intraerythrocytic parasites, *B. bovis* CE11/p2xHA-glmS-gfp-bsd parasites were exposed to 50 nM trtE or DMSO for 24 hours then fixed and processed for IFA. This *B. bovis* line expresses a soluble green fluorescent protein-blasticidin deaminase fusion protein (gfp-bsd),that was labeled with anti-GFP antibodies to visualize the parasite cytoplasm (Fig 3E, green fluorescence). Newly invaded piroplasms and mature meronts treated with DMSO showed normal morphology (Fig 3E upper panel, and S12 Fig, panels A and B). TrtE treated parasites showed a distinctly rounded morphology (Fig 3E, trtE, lower panel and S12 Fig, panels C and D) rather than the stereotypical pyriform shape (Fig 3E, upper panel and S12 Fig, panels A and B). Mis-localization of both the soluble gfp-bsd fusion protein (green label) and the RAP-1 protein (red label) was also observed. Normally, RAP-1 is associated with the rhoptries and secreted during invasion, often labeling a discrete area of the infected red blood cell (IRBC) membrane but not the IRBC cytoplasm (Fig 3E, upper panel and S12 Fig, panels A and B) [39]. Exposure to trtE resulted in diffuse distribution of RAP1 throughout the IRBC volume (Fig 3E, trtE, lower panel and S12 Fig, panels C and D), with what appeared to be persistent production and secretion during parasite development. Parasites exhibiting impaired cytokinesis were observed frequently (Fig 3E, trtE, lower panel, left image and S12 Fig, panel C).

## TrtE is highly selective for parasites over host cells

To evaluate the toxicity of trtE for host cells (HFF, HCT-8 and bovine turbinate cells, the host cells for *Sarcocystis*) at 20–30% confluency were treated for 24 hours with concentrations of trtE up to 24.5 μM. Toxicity was evaluated by measuring cellular ATP concentrations. The median toxic concentrations ($TC_{50}$) of trtE for host cells ranged from 16.8 μM to 6.3 μM (S13 Fig and Table 1) giving a selectivity indices of 1302 to 2633 (Table 1).

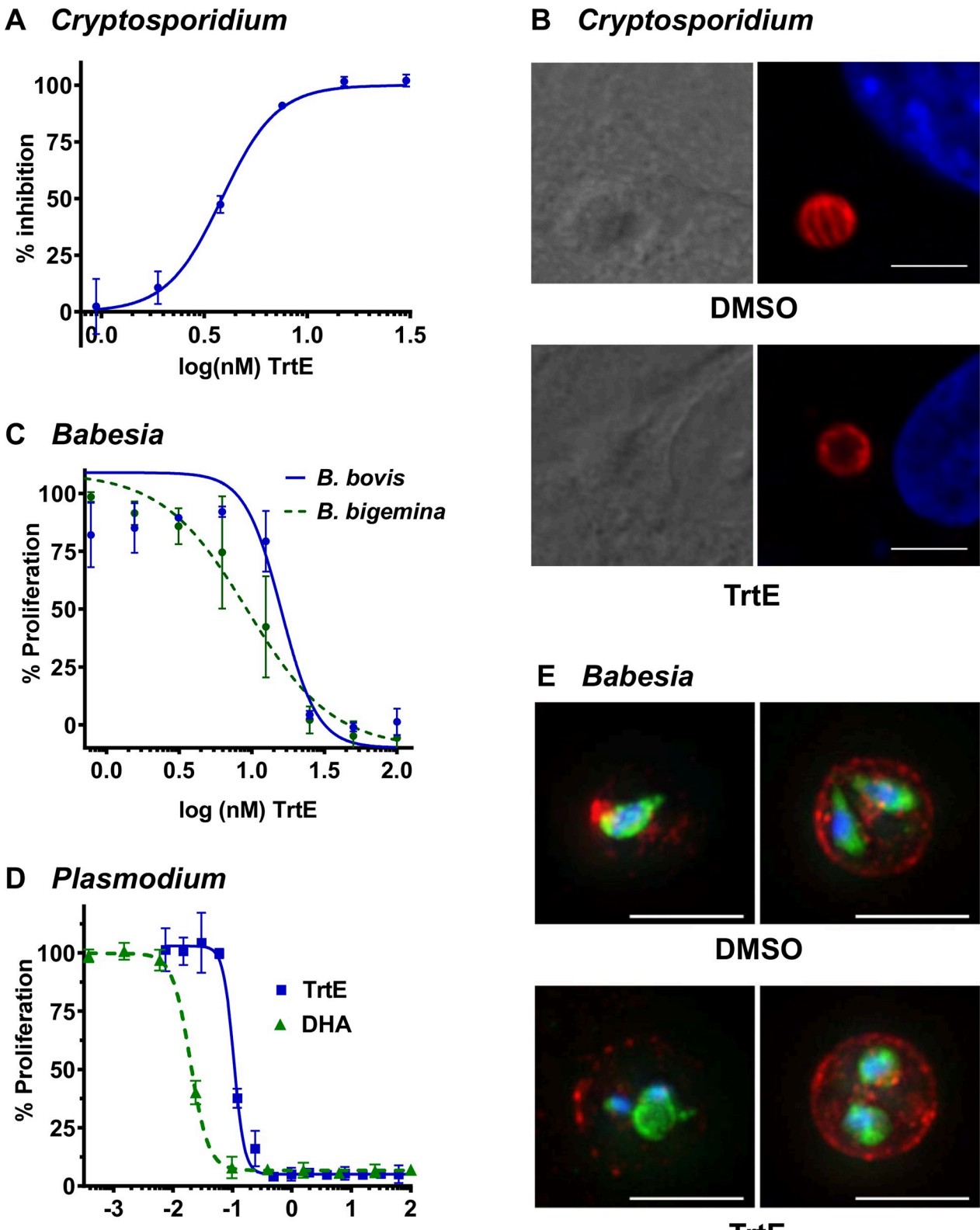

**Fig 3. Tartrolon E is active against a broad spectrum of apicomplexan parasites in vitro**: **A.** HCT-8 cells were infected with nanoluciferase expressing-*Cryptosporidium parvum* oocysts for 24 hours at which time dilutions of trtE were added to the infected cells. Parasite proliferation was

evaluated 48 hours post-treatment by quantification of nanoluciferase expression. The data was analyzed as described in the legend for Fig 2C; $EC_{50}$ = 3.85 nM with a 95% CI of 3.6–4.2 nM. **B.** *C. parvum* was allowed to infect HCT8 cells for 8 hours, then the infected cells were treated with 60 nM trtE or DMSO vehicle control for 12 hours. Infected cells were then fixed and processed for IFA. Images on left are DIC, images on right show corresponding IFA (Red: anti-gp15 antibody, blue: DAPI; scale = 5μm). Images shown are enlarged from S10 Fig, panels C and F. DMSO treated parasites formed normal type I meronts (**B**, upper panel and S10 Fig, panels A-C). There were few discernable parasites in trtE samples, and those parasite vacuoles that could be identified with anti-gp15 antibody did not appear to contain merozoites (**B.**, lower panel and S10 Fig, panels D-F).; **C, D.** *Babesia bovis*, *B. bigemina* (**C**) and *Plasmodium falciparum* (**D**) -infected erythrocytes were incubated with trtE or DMSO vehicle control and proliferation was measured after 72 hours by SYBR green incorporation. Percent proliferation was calculated relative to DMSO treated controls. $EC_{50}$s were calculated as described in the legend for Fig 2C. (**C**) *B. bovis*: $EC_{50}$ = 15.9 nM (95% CI 10–21 nM) and *B. bigemina*: $EC_{50}$ = 9.3 nM (95% CI 7.3–11.8 nM) (**D**) *P. falciparum*: $EC_{50}$ = 105 pM (95% CI 101–110 pM). The antimalarial drug dihydroartemesinin (DHA) is included for comparison, $EC_{50}$ = 20 pM (95% CI 19–21 pM). **E.** *B. bovis* CE11/p2xHA-glmS-gfp-bsd parasites were treated with DMSO (top images) or 50 nM trtE (bottom images) for 24 hours prior to fixation and immunostaining. The infected erythrocytes were labeled with rabbit anti-GFP (green) to visualize the parasite cytoplasm and an anti *B. bovis* RAP-1 mAb (red). Nuclei are visualized with DAPI. Images of DMSO treated parasites (top panels) show an early invaded erythrocyte (left) and mature meront (right), whereas images of trtE-treated parasites show divided meronts. Scale bars = 5 μm. Phase contrast and IFA-phase overlays of these images are shown in S12 Fig.

## TrtE is effective in vivo against the gastrointestinal parasite, *C. parvum*

Of all apicomplexans that infect both humans and domesticated animals, *Cryptosporidium* is a particular challenge for human and veterinary medicine as there is currently no effective treatment for this parasite. We chose to evaluate the in vivo efficacy of trtE using a neonatal mouse model of *C. parvum* infection [40]. This cost-effective model has been shown to have positive predictive value for justifying anti-cryptosporidial efficacy testing in the neonatal calf model [41,42]. In a preliminary experiment, ten, 8 day old mice were infected with *C. parvum* and then treated with four doses of 2 mg/kg of trtE (purification method 1) given orally every 12 hours for 2 days. Control mice received DMSO vehicle control. Five days post infection, mice were euthanized and the intestines processed for histology. Infection was scored by visual evaluation of infection intensity throughout the intestine by an investigator blinded to treatment. Since trtE significantly reduced the number of parasites found in the intestine of infected mice (Fig 4A–4C), the experiment was repeated with three concentrations of trtE (purification method 2). Eight-day-old mice, infected with *C. parvum*, were treated orally twice daily with trtE at 0.4, 4 or 40 mg/kg beginning two days post-infection and continuing for 2 days. On day 5, mice were euthanized and intestines processed for quantitative PCR to quantify infection [43]. At 4 mg/kg intestinal *Cryptosporidium* infection was reduced by 85% and was nearly eliminated at 40 mg/kg (Fig 4D). There were no overt signs of toxicity observed in any of the trtE treated mice.

## Discussion

To our knowledge, these studies are the first to identify an anti-apicomplexan compound with potent activity against parasites representing all branches of the apicomplexan phylogenetic tree. TrtE kills *T. gondii* parasites remarkably quickly at nanomolar concentrations, is highly

**Table 1. $EC_{50}$s, $TC_{50}$s and selectivity indices for parasites and their host cells.**

| Parasite | $EC_{50}$ (nM) | Host Cell | $TC_{50}$ (μM) | Selectivity Index |
|---|---|---|---|---|
| *T. gondii* | 3 | Human Foreskin fibroblasts | 7.9 | 2633 |
| *C. parvum* | 3.85 | HCT-8 colonic cancer cells | 6.3 | 1636 |
| *S. neurona* | 12.9 | Bovine turbinate cells | 16.8 | 1302 |
| *B. bovis* | 15.9 | | | |
| *B. bigemina* | 9.3 | | | |
| *T. equi* | 0.4 | | | |
| *P. falciparum* | 0.1 | | | |

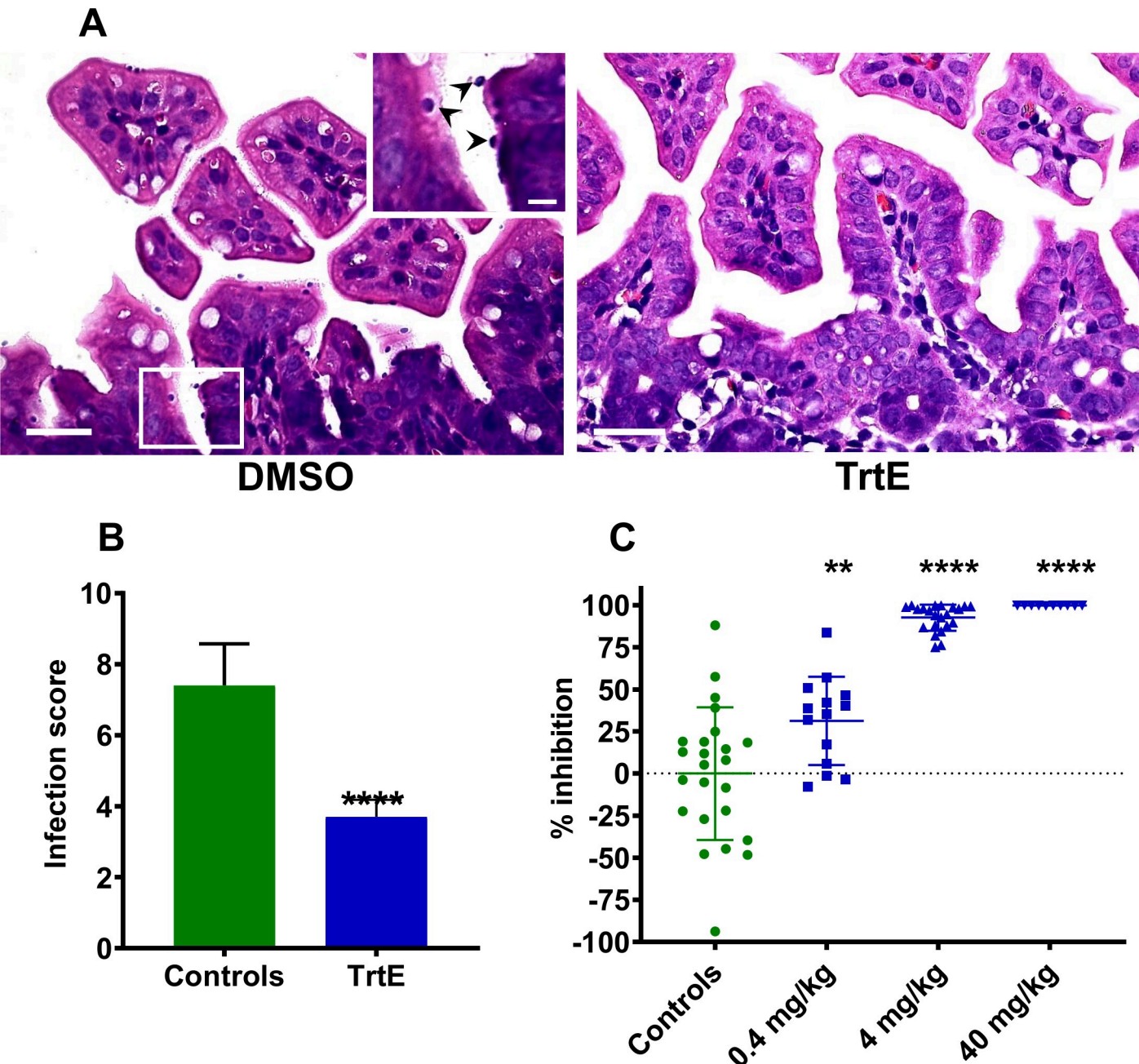

**Fig 4. Tartrolon E is effective against *Cryptosporidium parvum* infection in neonatal mice. A, B**. 8-day-old mice were infected with $5x10^4$ *C. parvum* oocsts. 48 hours post infection, mice were treated orally, every 12 hours, for 2 days with 2 mg/kg trtE (purification method 1). 5 days post infection mice were euthanized, and intestines processed for histology. **A. Left panel:** Section of ileum from a control mouse, scale bar = 25 μm. Area in box is enlarged in the upper right corner. Parasites are indicated with arrowheads. Scale bar = 5 μm. **Right panel**: Section of ileum from a mouse treated with trtE. No parasites were observed in this section. Scale bar = 25 μm. **B.** Slides with histological sections of the intestine from DMSO and trtE treated mice were generated and infection intensity was scored in the jejunum, ileum, caecum and colon by an investigator blinded to sample treatment. Scores for each section of the intestine were added together to give a total infection score for each animal. TrtE significantly reduced the number of parasites in the intestine (Mann-Whitney non-parametric t-test; ****p<0.0001). **C.** Neonatal mice were infected and treated as described for the first experiment but were dosed with three concentrations of trtE (purification method 2). Infection was evaluated by quantitative real-time PCR of intestinal tissue. Comparisons between control and experimental groups were made using a one-way ANOVA with Dunnett's Multiple Comparisons Test in Graphpad Prism. **p = 0.0023, ****p<0.0001.

specific for parasites over their host cells and is effective in vivo against *C. parvum* infection. The fact that this compound is produced by a mollusk symbiotic bacterium suggests that

efforts to identify new anti-parasitic drugs should focus on this potentially rich source. Symbiotic bacteria have lived in association with their host animals for millions of years, optimizing the production of compounds and enzymes that support and protect their host animal [44]. In contrast to compounds identified from microbes residing in soil or sediment, these molecules have the added advantage of having been developed within a metazoan host and may have better selectivity indices. As is the case with the shipworm and trtE, the exact role(s) that symbiont compounds play in the symbiosis is often unknown [45]. TrtE is produced in vivo in shipworms by the symbionts, and is thought to keep the digestive tract free of competing bacteria [18]. To achieve this, it has been proposed that the compound is transported from the site of production on the gill surface to the caecum, along with digestive enzymes. Consistent with this hypothesis, it has been reported that trtE is active against Gram positive and Gram negative bacteria, albeit at μM concentrations [18]. We postulate that trtE's potent anti-apicomplexan activity may have evolved from the symbiont's need to protect its host from gregarines, ubiquitous apicomplexan parasites of marine and terrestrial invertebrates (*Cryptosporidium* being a notable exception). Gregarines interact with their invertebrate hosts in ways ranging from beneficial to pathogenic [23] and to date there have been no reports of a shipworm-gregarine interaction, so this explanation is entirely speculative. However, the compound's pan anti-apicomplexan activity is consistent with the idea that the intended target is a member of this most basal apicomplexan subclass. Identification of the trtE molecular target will be very informative in this regard.

There are four other tartrolons that have been previously described, in addition to several other boron-containing macrodiolides [46–52]. These compounds are all chemically related to each other, in that they all contain symmetrical or semisymmetrical dimeric esters. Each monomeric unit contains two vicinal hydroxyl groups, a pyran moiety, and other structural features that are responsible for the three-dimensional structure of the dimer and the reversible complexation of the compounds with boron. Otherwise, structural details including methylation, ring size, oxidation state, and presence of additional rings, are quite variable across the class. Tartrolons A and B (identical to each other except that TrtA lacks a coordinated boron) were isolated from a cellulose degrading myxobacterium and found to have activity against Gram positive, but not Gram negative bacteria [46]. In a recent study, Tartrolon B, but not A, was shown to activate the NLRP3 inflammasome through its activity as a potassium ionophore [51]. Tartrolon C, produced by *Streptomyces* sp, was isolated on the basis of its insecticide activity [47], and was also found to inhibit hypoxia-inducible factor-1 transcription under hypoxic conditions [52]. Tartrolon D, (identical to trtE but again lacking a coordinated boron), was identified in a screen of marine actinomycetes for antitumor compounds [50]. All of these activities attributed to tartrolons A-D are observed at concentrations 100–1000 fold higher than the concentrations at which trtE demonstrates anti-parasitic activity. Thus, revisiting these compounds with an eye towards anti-apicomplexan activity could prove fruitful. Such investigations are further encouraged by the observation that two other boron containing macrodiolides, boromycin [48] and aplasmomycin [49], are active in vivo against poultry coccidiosis and rodent malaria, respectively.

Identification of trtE's mechanism of action, which could provide clues to its role in the shipworm symbiosis as well as point to new anti-parasitic drug targets, has proven elusive. The structurally related drugs, boromycin and TrtB, are potassium ionophores [51,53], and it would be expected that trtE would also have potassium ionophore activity. Ionophores have been used for years as feed additives to prevent coccidiosis in poultry but recently there has been renewed interest in compounds with this activity due to the potent anti-bacterial [53,54], anti-parasitic [55] and anti-cancer [56] activities exhibited by some of these compounds. However, it is unclear if the mode of action of these compounds is always attributable to their

ionophore activity. For example, monensin and salinomycin arrest *T. gondii* in S-phase and this cell cycle disruption is tied to activation of the DNA repair gene TgMSH1 [57] through an unknown mechanism. The transcriptome of monensin-treated *T. gondii* showed significant upregulation of histones and other genes associated with cell cycle arrest [57]. In contrast, the transcriptome of trtE treated *T. gondii* [58] did not show any evidence of cell cycle arrest. This observation suggests that the mechanism of action of trtE anti-parasitic activity is distinct from that of anti-parasitic ionophores currently in use. In fact, boromycin was shown to abrogate the G2 checkpoint in cancer cells treated with a DNA damaging agent, forcing the cells through the cell cycle [59].

Many questions regarding trtE's suitability for clinical development as yet remain unanswered. Production of gram quantities of the compound is feasible as trtE is the major compound isolated from *T. turnerae* T7901 cell pellets, the average yield is consistently 10 mg/L, and over time we have produced over 1.5 grams for experimental studies. Strain improvement methods could be employed to increase the yield to the levels needed for large animal testing and commercial production. We currently have no data on toxicity of trtE in animals besides the observation that neonatal mice receiving four doses of 40 mg/ml over two days did not exhibit overt signs of toxicity. TrtE is an ionophore and this class of compounds are reported to be highly toxic, which has limited their use in humans. However, there is a recent resurgence of interest in these compounds, not only for their antimicrobial properties [54], but also for their efficacy against multi-drug resistant cancers and cancer stem cells [56]. Because of this new interest in ionophores, there are ongoing efforts to create less toxic derivatives [56] or to reduce toxicity through alternate methods of delivery such as by the use of nanoparticles [60] or liposomal formulations [61,62]. Lessons learned from these studies may prove applicable to development of trtE as an effective therapeutic for parasitic diseases.

Likewise, the pharmacokinetic properties of trtE have yet to be characterized. While rapid absorption from the gut to the systemic circulation would be desirable for the treatment of the hemoparasites, effective treatment of *Cryptosporidium* is achieved with compounds that remain in the gut [63]. Here again, data on ionophore derivatives and delivery methods arising from the cancer treatment field may inform development of trtE formulations for treatment of various parasitic diseases.

Shipworms and their symbionts have recently gained considerable attention for their highly unusual lifestyles from their system of wood digestion [2] to the discovery that some shipworms derive their energy from sulfur-oxidizing bacteria that are related to symbionts of hydrothermal vent gastropods [64,65]. There are shipworms that make their homes in sea grass [66,67] and even limestone [68]. This study adds to the growing list of unexpected facets of shipworm symbiont biology and for the first time shows that these curious lifestyles have resulted in the development of metabolites with potential, yet largely untapped, medical applications. Remarkably, the anti-parasitic activity of trtE was discovered after screening only two strains of shipworm symbionts. There are numerous shipworm symbiont strains and species [69] with genomes rich in secondary metabolite loci [1]; we have thus only scraped the surface of this promising source of new therapeutics.

## Methods

### Growth of *T. turnerae*

*T. turnerae* T7901 [1] and T7902 [24] were streaked onto agar plates made with shipworm bacteria medium (SBM) containing 0.2% Sigmacell cellulose and 1% Bacto agar [24,25]. Individual colonies were expanded in SBM containing either 0.5% sucrose or 1% xylose as a carbon source. Cultures were grown at 30–35°C 35 rpm, in low form culture flasks until biofilms

formed. Cultures were centrifuged at 3500g for 30 minutes and the culture supernatant and pellet collected and stored at -80˚C. Cultures were periodically checked by PCR and sequencing of the 16S rRNA gene [69].

## Bioassay-guided fractionation

*T. turnerae* T7901 cells were separated from media by centrifugation. The media (100 ml) was extracted three times with ethyl acetate (75 ml). The ethyl acetate fraction was dried under vacuum to yield an extract (25 mg). This extract was dissolved in methanol (2 ml), which was loaded onto end-capped C18 resin (2 g). The loaded resin was dried under vacuum and loaded onto a column containing additional C18 resin (5 g). The column was washed with water (50 ml), then fractions were eluted using a methanol / water step gradient (60%, 70%, 80%, 90% and 100% methanol; 100 mL each), to provide fractions 1–7, which were dried under vacuum. The seven fractions were re-suspended in DMSO to 1 mg/ml and further diluted in cell culture medium for bioassay testing that included monolayer protection assays and immunofluorescence assays. HFF cells, seeded into 96-well plates, were infected with *T. gondii* RH strain for 24 hours, at which time dilutions of each fraction were added to the plate. After 48 hours of incubation, the cells were fixed and stained with trypan blue to assess monolayer damage.

## Purification of trtE from *T. turnerae* T7901, method 1

To obtain pure trtE, the cell pellet from 1 liter of *T. turnerae* T7901 culture was extracted three times with acetone (100 ml). The solution was dried under vacuum to yield an extract (70 mg), which was dissolved in methanol (2 ml) and flushed through a cartridge containing end-capped C18 resin. The flow-through was purified by HPLC using 85% acetonitrile in water to obtain trtE (3.4 mg). A Phenomenex Onyx Monolithic semiprep C18 column (100 x 10 mm) was used for HPLC, as conducted on a Hitachi Elite Lachrom System equipped with a Diode Array L-2455 detector. LC/ESI-MS was performed using a Micromass Quattro-II (Waters) instrument on an analytical Agilent Eclipse XDB-C18 column (4.6 x 50 mm, 5 μm) with a linear gradient of 1%–99% B over 20 min, where the mobile phase consisted of solvent A ($H_2O$ with 0.05% formic acid) and solvent B (acetonitrile).

## Purification of trtE from *T. turnerae* T7901, method 2

A cell pellet resulting from 1.2 liters of T7901 culture was extracted with acetone (3 x 300 ml) with vortexing and sonication, and the solvent was evaporated under reduced pressure to afford an extract (149.2 mg). The extract was solubilized in methanol (2 ml) and the material was subjected to solid phase extraction (SPE) (Waters Sep-Pak Vac C18 cartridge, 35 cc (10 g)). The SPE cartridge was equilibrated with 10% MeCN/$H_2O$, and after loading the solvated extract, step gradient elution was performed by flushing the cartridge with 2 x 25 mL of the following MeCN/$H_2O$ solvent mixtures: 10% (fractions 1 and 2), 25% (fractions 3 and 4), 50% (fractions 5 and 6), 75% (fractions 7 and 8), 100% (fractions 9 and 10), and the cartridge was then flushed with acetone (4 x 25 ml, fractions 11–14). trtE was observed in fractions 10, 11, and 12. These fractions were consolidated and the solvent was evaporated under reduced pressure to afford a light yellow solid (35.5 mg). This material was then solvated in methanol (3.5 ml) and passed through a syringe filter (0.2 μm) to remove particulate matter. The flow-through was subjected to semi-preparative HPLC on an Agilent Series 1100/1200 HPLC System equipped with a photo-diode array detector (DAD) using a Phenomenex Synergi™ 4 μm Hydro-RP 80 Å column (250 x 10 mm) employing isocratic elution with 95% MeCN/$H_2O$ at a flow rate of 3 mL/min to obtain trtE as a white solid (3.5 mg, $t_R$ = 29.3 min).

The sample was analyzed via LC-MS and $^1$H NMR. and then was dried and placed under high vacuum overnight before use in biological testing. LC-MS was performed on an Agilent Series 1200 HPLC System equipped with a photo-diode array detector (DAD) and a 6130 quadrupole mass spectrometer using an analytical Phenomenex Luna 5 μm C18(2) 100 Å column (100 x 4.6 mm) with a linear gradient of 45%– 90% MeCN + 0.1% formic acid (FA) / H2O + 0.1% FA over 20 min, then linear gradient to 100% MeCN + 0.1% FA over 1 min, then isocratic MeCN + 0.1% FA for 7 min, then linear gradient to 45% MeCN + 0.1% FA over 1 min, then isocratic 45% MeCN + 0.1% FA for 6 min all at a flow rate of 0.7 mL/min. Under these LC-MS conditions, trtE was found to have $t_R$ = 20.1 min. Comparison of the $^1$H NMR spectrum of the product isolated through Method 2 (S9 Fig) and the $^1$H NMR spectrum of the product isolated through Method 1 (S6 Fig) indicates that the compound is trtE.

For $^1$H NMR (500 MHz, CD$_3$OD): δ = 6.12–6.02 (4H, m), 5.88 (2H, dt, $J$ = 14.4, 4.6 Hz), 5.30–5.25 (2H, m), 4.72–4.66 (2H, m), 4.49 (2H, s), 4.44–4.40 (2H, m), 3.99–3.95 (2H, m), 3.21 (4H, dd, $J$ = 18.3, 10.4 Hz), 2.63 (2H, dd, $J$ = 14.0, 3.9 Hz), 2.49–2.44 (4H, m), 2.42–2.33 (2H, m), 2.16–2.11 (2H, m), 2.04–1.90 (4H, m), 1.86–1.79 (2H, m), 1.78–1.65 (4H, m), 1.64–1.58 (4H, m), 1.44–1.38 (2H, m), 1.34–1.23 (4H, m), 1.16 (6H, d, $J$ = 6.1 Hz), 1.05 (6H, d, $J$ = 6.5 Hz).

## Immunofluorescence assays (IFAs)

For all IFAs, the investigator reading the slides and collecting images was blind to the sample treatment. Host cells (human foreskin fibroblast (HFF) cells for *T. gondii* and HCT-8 for *C. parvum*) were seeded onto coverslips or 8 well chamber slides and allowed to establish monolayers. Host cells were infected for 8 to 24 hours at which time bacterial culture supernatants, C18 column fractions, trtE or DMSO (vehicle control) were added. 24 hours post treatment, infected cells were fixed with methanol, and parasites detected with rabbit anti-SAG1 antibody for *T. gondii* or rabbit anti-gp15 for *C. parvum* as we have described previously [70,71]. *B. bovis* parasite line CE11/p2xHA-glmS-gfp-bsd was maintained in in vitro culture under microaerophilic conditions [72]. This CE11 line is transiently transfected with a plasmid that expresses nanoluciferase from a *B. divergens* promoter and green fluorescence protein-blastici-din (gfp-bsd) from the selection cassette. For immunofluorescence, cultured parasitized erythrocytes were suspended in culture medium containing either 1.0% (v/v) dimethyl sulfoxide (DMSO/CM) or 100 nM trtE in DMSO/CM, then were placed under culture conditions for 24h. Cells were fixed as described [73] followed by permeabilization with ice-cold 0.01% Nonidet P-40 in phosphate buffered saline (PBS). Fixative was killed with 65 mM Tris-HCl, 100 mM NaCl, pH 7.4 (TBS) for 30 minutes, then cells were resuspended in 5 mM CuSO4 in TBS for 90 minutes at room temperature to reduce hemoglobin autofluorescence [74]. Cells were blocked overnight in 3% (w/v) bovine serum albumin in PBS prior to antibody labeling. Cells were incubated with rabbit anti-GFP (GenScript; Piscataway, NJ; cat. #A01388) and mouse monoclonal MBOC79B1 (a gift from G. Palmer) anti-BbRAP-1 protein antibodies, and reactions detected with goat anti-rabbit IgG-Alexafluor 488 and goat anti-mouse IgG-Alexafluor 594 antibodies (Invitrogen; Carlsbad, CA). After washing, cells were counter-stained with 4′,6-Diamidine-2′-phenylindole dihydrochloride (DAPI; Sigma Chemical; St. Louis, MO) mounted with 1,4-diazabicyclo-(2,2,2)-octane anti-fade (DABCO; [75]). Images were captured with an Olympus BX50 microscope fitted with a 100x oil-immersion (N.A. 1.35) phase-contrast objective, using IP Labs software (Scanalytics, Inc.; Rockville, MD). Images were deconvolved using an iterative Weiner filter preconditioned Landweber method, as implemented for ImageJ [76]. Mean background levels were determined for individual images by assaying areas lacking cells, and the values were subtracted, whereas maximum fluorescent values were individually set as the upper limit. The three fluorophore channels were merged using the RGB

gray merge plug-in. All image processing was performed with ImageJ 1.51j8 (Wane Rasband; http://imagej.nih.gov/ij).

## Parasite inhibitor response assays

Unless otherwise noted all samples were run in triplicate and assays were conducted a minimum of three times. In all assays, inhibition of growth in the presence of trtE was determined by comparison to growth in the presence of the vehicle control (DMSO).

**T. gondii.** *T. gondii* parasite lines were maintained by passage in HFFs as previously described [77]. To determine the effect of trtE on intracellular parasites, 96 well plates containing confluent HFFs were infected with *T. gondii* strain ME49 modified to express green fluorescent protein (GFP) and luciferase (Luc), generated as described [33] or *T. gondii* RH strain GFP/Luc (a kind gift from Dr. Jeroen Saeij, UC Davis). 24 hours post infection, dilutions of trtE or DMSO were added to the wells and infection allowed to proceed for another 24 hours. Growth was monitored by luciferase expression using Bright-glo Luciferase assay system (Promega Corp., Madison, WI). To determine the length of treatment needed to kill intracellular parasites, trtE was added to infected cells and then the compound removed 0.5, 1, 2 and 4 hours later by washing the monolayer three times with complete media. Infected cells were then incubated for 72 hours before growth was quantified by luciferase expression. For time-lapse images, HFFs were seeded into 35 mm glass bottom microwell dishes (MatTek Corporation, Ashland, MA). Cells were infected with *T. gondii* RH strain GFP/Luc for 24 hours at which time 24 nM trtE or an equivalent dilution of DMSO was added to the infected cells. Beginning one hour post-treatment phase contrast images were collected every 5 minutes for 12 hours with an Olympus VivaView FL incubator microscope and MetaMorph software (Olympus Life Sciences, Waltham, MA). Images were processed into a video using Image J.

To determine the direct effect of trtE on *T. gondii*, RH GFP/Luc tachyzoites that had exited from host cells were combined with trtE at varying concentrations and incubated at 37˚C and 5% CO2 for 2 hours. Tachyzoites in 0.01% DMSO were run in parallel. The parasites were washed to remove compound and added to 96-well plates containing confluent HFFs. Growth was monitored 24 hours later using Bright-glo Luciferase assay system (Promega). Percent inhibition was calculated as ((luciferase reading DMSO treated parasites—luciferase reading of trtE treated parasites)/ luciferase reading DMSO treated parasites)*100.

**C. parvum.** *C. parvum* oocysts expressing nanoluciferase (nLuc) were obtained from the *Cryptosporidium* Production Laboratory at the University of Arizona and inhibitor response assays were conducted as previously described [30,63].

**Sarcocystis neurona.** Culture of *Sarcocystis neurona* GFP/Luc merozoites in bovine turbinate cells and in vitro drug response assays were conducted as previously described [33].

**Plasmodium falciparum.** The maintenance of *Plasmodium falciparum* cultures and conducting of in vitro inhibitor response assays have been previously described [34,35].

**Theileria equi.** The effect of trtE on proliferation of *T. equi* was evaluated in a 72 hr *in vitro* parasite growth inhibition assay performed as previously described [36] using the USDA Florida strain of *T. equi* [78].

**Babesia species.** *B. bovis* and *B. bigemina* inhibitor response assays were adapted from components of [79–81]. *B. bovis* and *B. bigemina* parasites were maintained in bovine erythrocytes under microaerophilic stationary phase conditions in vitro as described elsewhere [72,82]. Prior to use in growth studies parasites were grown in erythrocytes from which leukocytes were depleted [83]. Cultures were set up at 0.20% parasitized erythrocytes, in quadruplicate, in complete medium supplemented with trtE. Final concentration of DMSO in all cultures was 1%, and was consistent throughout. Following 72h growth under 90% nitrogen,

5% oxygen, 5% carbon dioxide cells were harvested by thorough resuspension, and transferred to a black, opaque 96-well plate. Cells were freeze-thawed at -80˚C overnight, then mixed with an equal volume of SYBR Green lysis buffer. Lysis buffer was composed of 20 mM Tris, pH 7.5, 5 mM EDTA, 0.008% (w/v) saponin, 0.08% (v/v) NP40, and contained 0.2 μl 10,000x SYBR Green I dye ml$^{-1}$ (S7563; Invitrogen; Carlsbad, CA). Plates were incubated for one hour at room temperature in the dark prior to reading in a microplate reader, at 485 nm excitation and 528 nm emission wavelengths.

## Cytotoxicity of trtE for host cells

HCT8, HFF and BT cells were seeded in 96-well plates to achieve 20–30% confluence after 24 hours of growth. Dilutions of trtE or DMSO were added to the wells and proliferation of the cells evaluated 24 hours later by quantifying cellular ATP levels with CellTiter-Glo Cell viability assay (Promega).

## Neonatal mouse model of *Cryptosporidium parvum* infection

8-day-old mice were infected with $5\times10^4$ oocysts by oral gavage. In the initial preliminary experiment using trtE purified by method 1, mice were treated 48 hrs post infection with 2 mg/kg trtE every 12 hours by oral gavage for a total of 4 treatments. Control mice received vehicle control (DMSO diluted in DMEM containing 10% fetal calf serum). Mice were sacrificed 5 days post infection and intestines prepared for histological evaluation. Slides were blinded and the intensity of infection scored in the terminal jejunum, ileum, caecum and colon as previously described [40]. Infection intensity scores for each region were added together to give a total infection score for the animal.

In the second set of experiments, mice were infected and treated as before, with 0.4, 4 or 40 mg/kg of trtE purified using method 2. Control mice received vehicle control in parallel. Mice were sacrificed 5 days post-infection and infection quantified by qPCR as previously described [42,43]. The experiment was conducted three times.

## Statistical analysis

All statistical analysis was done in GraphPad Prism (San Diego, CA). EC$_{50}$s or TC$_{50}$s were calculated using the log[inhibitor]vs response-variable slope (four parameter) regression equation and inhibition curves were compared using the Extra sum-of-squares F test. For the preliminary mouse experiment, treated and control groups were compared using the Mann-Whitney non-parametric t-test. For subsequent mouse experiments, the data from all mice in the three replicate experiments were pooled and the data analyzed by a one way ANOVA with Dunnett's Multiple Comparisons test to compare differences between the groups.

## Ethics statement

All animal studies described in this manuscript were approved by the University of Arizona IACUC (protocol # 09–120). Animals were euthanized by isoflurane inhalation followed by cervical dislocation as is consistent with the current AVMA Guidelines for the Euthanasia of Animals.

## Supporting information

**S1 Fig. *T. gondii* treated with culture supernatants from shipworm symbionts T7902 and T7901.** HFF cells were infected with *T. gondii* parasites for 24 hours then medium containing bacterial culture supernatants at final dilutions of 1:100 or 1:250 were added to the infected

cells. 24 hours post-treatment cells were fixed and processed for IFAs. Parasites were labeled with rabbit anti-SAG1 antibody detected with Alexafluor 594-labelled goat anti-rabbit IgG (green). Host cell nuclei are visualized with DAPI. Scale bar = 10 μm.
(DOCX)

**S2 Fig.** Bioassay-guided fractionation of T. turnerae T7901 culture supernatant: A. Methanol/water fractions eluted off a C18 column were lyophilized, resuspended in DMSO and tested for anti-Toxoplasma activity using a monolayer protection assay. Details of the extractions can be found in methods. For the monolayer protection assay, HFF cells were seeded into 96 well plates, infected for 24 hours with T. gondii RH strain parasites, and then dilutions of extracts were added to the infected cells. 24 hours post-treatment the cell monolayers were fixed and stained with trypan blue. B. HFF cells infected with T. gondii RH strain parasites for 24 hours were treated with the 90% methanol fraction diluted to 10 μg/ml or DMSO control. 24 hours post treatment, infected cells were fixed and processed for IFAs. Parasites were labeled with rabbit anti-SAG1 antibody detected with Alexafluor 594-labelled goat anti-rabbit IgG (red). Host cell nuclei are visualized with DAPI.
(DOCX)

**S3 Fig. HPLC chromatogram of trtE purified by method 1, detected in 224 nm by DAD.**
The purity of trtE was calculated > 99%.
(DOCX)

**S4 Fig. Mass spectrometry data of trtE purified by method 1.**
(DOCX)

**S5 Fig. $^1$H NMR spectra in CDCl$_3$ of trtE purified by Method 1.**
(DOCX)

**S6 Fig. $^1$H NMR spectra in CD$_3$OD of trtE purified by Method 1.**
(DOCX)

**S7 Fig. Intracellular stages of *Toxoplasma gondii* treated with trtE.** *T. gondii* RH strain tachyzoites were allowed to infect HFF cells for 24 hours at which point trtE was added to a final concentration of 60nM. Cells were fixed and processed for IFA 24 hours after the addition of the compound. DMSO was run in parallel as a negative control. Images on the left are DIC, images on the right show the IFA. Parasites are labeled with rabbit anti-SAG1 antibody detected with Alexafluor594-labeled goat anti-rabbit IgG (red). Host cell nuclei are visualized with DAPI (blue). Panel **A** shows parasites treated with DMSO. Panels **B** through **F** show infected cells treated with trtE. Scale bar = 10μm.
(DOCX)

**S8 Fig. LC-MS data of trtE purified by Method 2.**
(DOCX)

**S9 Fig. $^1$H NMR spectra of trtE purified by Method 2 (500 MHz, CD$_3$OD).**
(DOCX)

**S10 Fig. Intracellular *C. parvum* parasites treated with trtE.** HCT-8 cells were infected with *C. parvum* oocysts for 8 hours, at which time cells were washed to remove extracellular parasites and medium containing 60 nM trtE or DMSO was added to the infected cells. 12 hours later, infected cells were fixed and processed for IFAs. Images on the left are DIC, images on the right show the IFA. Parasites are labeled with rabbit anti-gp15 antibody detected with Alexafluor594-labeled goat anti-rabbit IgG (red). Host cell nuclei are visualized with DAPI

(blue). Panels **A** through **C** show parasites treated with DMSO. Panels **D** through **F** show parasites treated with trtE. Very few discernable parasites could be found in the trtE treated cells. Scale bar = 5μm.
(DOCX)

**S11 Fig. TrtE exhibits broad spectrum anti-apicomplexan activity in vitro. A.** Bovine turbinate cells infected with luciferase expressing *Sarcocystis neurona* merozoites were treated with trtE for 24 hours and parasite growth evaluated by luciferase expression. $EC_{50}$s were determined using the log[inhibitor]vs response-Variable slope (four parameter) regression equation in Graphpad Prism, $EC_{50}$ = 12.9 nM with a 95%CI of 11–15 nM. **B**. *Theileria equi*-infected erythrocytes were incubated with trtE or DMSO vehicle control and proliferation was measured after 72 hours by SYBR green incorporation. Percent proliferation was calculated relative to DMSO treated controls. Inhibition of proliferation was analyzed as described for (**A**), $EC_{50}$ = 391 pM (95% CI 286–550 pM).
(DOCX)

**S12 Fig. *Babesia bovis*-infected erythrocytes treated with trtE.** *B. bovis* CE11/p2xHA-glmS-gfp-bsd parasites were treated with DMSO (top panel) or 50 nM trtE (bottom panel) for 24h prior to fixation and immunostaining. The infected erythrocytes were labeled with rabbit anti-GFP detected with goat anti-rabbit IgG (H&L chains)-Alexafluor 488 (green) to visualize the parasite cytoplasm and an anti *B. bovis* RAP-1 mouse mAb (MBOC79B1) detected with goat anti-mouse IgG (H&L chains)-Alexafluor 594 (red). Nuclei were counterstained with DAPI (blue). Left panels show the merger of the three color channels, middle panels show the fluorescence image overlaid the phase-contrast image and the right panels show the phase contrast image. Control panels show an early invaded erythrocyte (**A**) and mature meront (**B**), whereas the trtE-treated parasites (**C** and **D**) shown are divided meronts. Scale bars = 5 μm.
(DOCX)

**S13 Fig. Cytotoxicity of trtE for host cells.** Host cells were seeded into the wells of a 96 well plate to achieve approximately 25% confluency. 24 hours after seeding, trtE was added at the concentrations indicated. DMSO was run in parallel. 24 hours post addition of compound, viability of the cells was determined by quantification of ATP with CellTiter Glo. $TC_{50}$s were determined using the log[inhibitor]vs response-Variable slope (four parameter) regression equation in Graphpad Prism. **A**. HFF: $TC_{50}$ 7.9 μM (95% CI 6.1–10.4), **B**. HCT-8: $TC_{50}$ 6.3 μM (95% CI 5–8), **C**. BT cells: $TC_{50}$ 16.8 μM (95% CI 11.6–28.8).
(DOCX)

**S1 Movie. Time lapse video of the effects of trtE on intracellular *T. gondii*.**
(MP4)

**S2 Movie. Time lapse video of intracellular *T. gondii* treated with vehicle control (DMSO).**
(MP4)

## Acknowledgments

The authors thank Yu-Ping Xiao for excellent technical assistance.

## Author Contributions

**Conceptualization:** Roberta M. O'Connor.

**Formal analysis:** Roberta M. O'Connor.

**Funding acquisition:** Roberta M. O'Connor.

**Investigation:** Roberta M. O'Connor, Felix J. Nepveux V, Jaypee Abenoja, Gregory Bowden, Patricia Reis, Josiah Beaushaw, Rachel M. Bone Relat, Iwona Driskell, Fernanda Gimenez, Michael W. Riggs, Deborah A. Schaefer, Eric W. Schmidt, Zhenjian Lin, Timothy R. Ramadhar, David R. Allred, Pradipsinh Rathod, Laura Chery, John White.

**Methodology:** Gregory Bowden, Heather M. Fritz.

**Project administration:** Roberta M. O'Connor.

**Resources:** Daniel L. Distel, Jon Clardy, Timothy R. Ramadhar.

**Supervision:** Roberta M. O'Connor, Iwona Driskell, Michael W. Riggs, Deborah A. Schaefer, Eric W. Schmidt, Jon Clardy, David R. Allred, Pradipsinh Rathod.

**Validation:** Roberta M. O'Connor, Felix J. Nepveux V, Jaypee Abenoja, Gregory Bowden, Fernanda Gimenez, Deborah A. Schaefer, Eric W. Schmidt, Zhenjian Lin, Timothy R. Ramadhar, David R. Allred.

**Visualization:** Roberta M. O'Connor, Zhenjian Lin, Timothy R. Ramadhar, David R. Allred.

**Writing – original draft:** Roberta M. O'Connor, Fernanda Gimenez, Michael W. Riggs, Deborah A. Schaefer, Eric W. Schmidt, Zhenjian Lin, Timothy R. Ramadhar, David R. Allred, Laura Chery, John White.

**Writing – review & editing:** Roberta M. O'Connor, Fernanda Gimenez, Michael W. Riggs, Deborah A. Schaefer, Eric W. Schmidt, Zhenjian Lin, Daniel L. Distel, Timothy R. Ramadhar, David R. Allred, Heather M. Fritz, Laura Chery, John White.

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
