## [Decision Letter · Decision Letter 0]

26 Feb 2020

Dear Dr O'Connor,

Thank you very much for submitting your manuscript "A symbiotic bacterium of shipworms produces a compound with broad spectrum anti-apicomplexan activity." for consideration at PLOS Pathogens. As with all papers reviewed by the journal, your manuscript was reviewed by members of the editorial board and by several independent reviewers. In light of the reviews (below this email), we would like to invite the resubmission of a significantly-revised version that takes into account the reviewers' comments.

All reviewers agreed that this is an interesting study that brings a new line of compounds to the attention of the parasitology drug development community. There were however, multiple concerns that need to be addressed.

Please word your statements carefully. Fig. 1 e.g. claims “… supernatants exhibited parasiticidal activity that inhibited the growth of T. gondii”. So did they kill or just inhibit growth? In this experiments the authors only measure growth and thus should qualify their statement accordingly. If they stand by kill as a core to their claim, that can be tested in T. gondii using drug wash-out, lysis and plaque assay. Please comb the entire manuscript carefully to match claim with strength of data. Note that PLoS Pathogens typically does not re-review and will reject in cases where the standard is not met.

The reviewers had several question about how growth was quantified in particular for Cryptosporidium. One reviewer suggested the use of luciferase parasites, which the authors actually did. Part of the issue here may be the inconsistent labeling of the figures throughout which confused the reviewer and may well do so for readers, it distracted the editor. This is likely due to the fact that different co-authors prepared different panels, but “editorial cleanup” from the lead author could really help the clarity. Please settle on uniform fonts, sizes and colors. One reviewer suggested to clearly indicate the parasite species in each panel in a consistent fashion -- that is an excellent suggestion. Also take a careful look at the legends and make sure they give full detail. This is a relatively brief article, so there is room for clear description.

While the editor agrees with one of the reviewers that electron microscopy would give a higher resolution view of the morphological consequence of treatment he feels that that consequence is not really at the center of the manuscript. Inhibition is a the center, and that authors do show.

The reviewers have a number of important questions about toxicity and other aspects that may weigh in on the future use of these compounds including synthesis and isolation and pharmacokinetics. Please discuss these thoroughly, clarify your data as requested and supply further information on toxicity. Potential users of a cryptosporidiosis drug in particular (there is treatment for toxoplasmosis) will require a high safety margin, this is particularly true for pediatric applications.

Some reviewers were unable to evaluate the supplementary movie files. The editor notes that they were in gif format which can be viewed with web browser but is not the typical format standard used. Please confer with editorial office for guidance (most use mov or avi). Also a running clock or time stamp would help to appreciate the result of this experiment.

We cannot make any decision about publication until we have seen the revised manuscript and your response to the reviewers' comments. Your revised manuscript is also likely to be sent to reviewers for further evaluation.

Sincerely,

Boris Striepen

Guest Editor

PLOS Pathogens

Vern Carruthers

Section Editor

PLOS Pathogens

Kasturi Haldar

Editor-in-Chief

PLOS Pathogens

orcid.org/0000-0001-5065-158X

Michael Malim

Editor-in-Chief

PLOS Pathogens

orcid.org/0000-0002-7699-2064

Please word your statements carefully. Fig. 1 e.g. claims “… supernatants exhibited parasiticidal activity that inhibited the growth of T. gondii”. So did they kill or just inhibit growth? In this experiments the authors only measure growth and thus should qualify their statement accordingly. If they stand by kill as a core to their claim, that can be tested in T. gondii using drug wash-out, lysis and plaque assay. Please comb the entire manuscript carefully to match claim with strength of data. Note that PLoS Pathogens typically does not re-review and will reject in cases where the standard is not met.

The reviewers had several question about how growth was quantified in particular for Cryptosporidium. One reviewer suggested the use of luciferase parasites, which the authors actually did. Part of the issue here may be the inconsistent labeling of the figures throughout which confused the reviewer and may well do so for readers, it distracted the editor. This is likely due to the fact that different co-authors prepared different panels, but “editorial cleanup” from the lead author could really help the clarity. Please settle on uniform fonts, sizes and colors. One reviewer suggested to clearly indicate the parasite species in each panel in a consistent fashion -- that is an excellent suggestion. Also take a careful look at the legends and make sure they give full detail. This is a relatively brief article, so there is room for clear description.

While the editor agrees with one of the reviewers that electron microscopy would give a higher resolution view of the morphological consequence of treatment he feels that that consequence is not really at the center of the manuscript. Inhibition is a the center, and that authors do show.

The reviewers have a number of important questions about toxicity and other aspects that may weigh in on the future use of these compounds including synthesis and isolation and pharmacokinetics. Please discuss these thoroughly, clarify your data as requested and supply further information on toxicity. Potential users of a cryptosporidiosis drug in particular (there is treatment for toxoplasmosis) will require a high safety margin, this is particularly true for pediatric applications.

Some reviewers were unable to evaluate the supplementary movie files. The editor notes that they were in gif format which can be viewed with web browser but is not the typical format standard used. Please confer with editorial office for guidance (most use mov or avi). Also a running clock or time stamp would help to appreciate the result of this experiment.

Reviewer's Responses to Questions

**Part I - Summary**

Reviewer #1: This manuscript details the discovery of a new, highly effective anti-Apicomplexan compound that is produced by the symbiotic bacteria of shipworms. It encompasses a tremendous amount of work from multiple labs that is highly valuable information for the parasitology field. Overall, much of the data is excellent and exciting, especially the in vivo Cryptosporidium data. However, it is unclear if the immunofluorescent slides were blinded to reduce bias and the images need DIC or phase (as well as multiple examples in supplemental data) to support the phenotypes that they are claiming. Below are points that need to be addressed.

Reviewer #2: This is a very interesting paper looking in shipworm bacteria for natural products that could be used as antiparasitics for protozoan pathogens. The authors reasoned that shipworms are attacked by gregarine pathogens, which are related to apicomplexa parasites such as Toxoplasma, Cryptosporidium, Malaria, Babesia, Theileria, Neospora, and Sarcocystis. The screened two strains of commensulate bacteria known to inhabit shipworms and found that one of the two strains produced an extract that was very potent against the Toxoplasma gondii. They then used activity based purification and found that the natural product was Tartrolon E (TrtE), which had previously been described for its antibacterial activity in the micromolar activity. Yet the TrtE activity for the anti toxoplasma (and subsequently found anti- Crypto, Babesia, Neospora, Theileria, and Sarcocystis) was in the nanomolar to high picomolar level, thus >1000 more active against apicomplexan protozoan. There was activity of TrtE against mammalian cells, but generally not until above 10uM, suggesting promising lack of toxicity. The authors went on to show that TrtE delivered orally to newborn mice infected with C. parvum could greatly reduce the infection. The paper is very well written and the results are very exciting.

Reviewer #3: The manuscript by O’Connor and colleagues reports an interesting discovery of anti-apicomplexan activity of a polyketide compound produced by a shipworm symbiotic bacteria Teredinibacter turnerae.

Major findings include:

Discovery of bioactive compound:

T. turnerae (one of the two strains studied) secreted a compound that displayed anti-Toxoplasma activity, in which tartrolon E (TrtE) was confirmed to be responsible for the anti-parasitic activity.

In vitro activity on several apicomplexans:

TrtE was highly efficacious against Toxoplasma (3 strains), Babesia (2 species) and Cryptosporidium parvum in vitro (EC50 values at lower nM range).

TrtE could quickly inactivate Toxoplasma tachyzoites that were free in medium.

In vivo activity on Cryptosporidium:

TrtE was effective against cryptosporidial infection in vivo using a neonatal mouse model of

Overall, these findings are novel and significant in the discovery of new compounds against various apicomplexan parasites. Anti-parasitic activity of TrtE was new, although tartrolon ionophores were first discovered over 20 years ago, and TrtE was reported in 2013, but mainly for their antibacterial activities.

**Part II – Major Issues: Key Experiments Required for Acceptance**

Reviewer #1: 1) Immunofluorescent images are not a good morphological tool, especially when the authors are showing only one image with no DIC or phase. While it would be better to have EM, at the very least Figures 1b, 2b, 3b and 3e must have the DIC or phase controls, multiple images taken and added to supplemental data, and the slides need to be blinded before pictures are taken to reduce bias.

2) For the extracellular Toxoplasma time course experiment (Fig 2e), it is unclear if no drug extracellular controls were done for the same time periods because Toxoplasma becomes rapidly unhealthy during extracellular incubations.

3) No direct evidence of Cryptosporidium growth arrest in tissue culture, as the immunofluorescence is not high quality. Isn’t there a luciferase containing Cryptosporidium strain? Why was that not used so that the tissue culture data could be quantitative. Otherwise, could the slides be blinded and the authors measure vacuole size, or could they do the qPCR?

Reviewer #2: None

Reviewer #3: Toxicity to animals needs to be more thoroughly evaluated, as this is the major issue in developing ionophores into therapeutics.

To support TrtE as a "broad-spectrum" anti-apicomplexan compounds, in vivo activity against at least one more parasite may be needed.

In mouse model cryptosporidial infection model, what were the oocyst counts in feces?

**Part III – Minor Issues: Editorial and Data Presentation Modifications**

Reviewer #1: 1) The results need more methods added so the reader can follow the experiments. For example, lines 149-156 are very confusing and it would help to add detail such as grown for 24 hours, then add drug for 24 hours.

2) The movies in supplemental were not functioning.

3) Figure 3a, c and d should be labeled with the parasite name so the reader can more easily follow this complicated figure.

Reviewer #2: 1) Did you collect plasma or GI tissues during mouse therapy and try to measure TrtE levels? As a point of reference, it would be nice to know if there was systemic exposure of TrtE. This could help guide route of administration for other systemic parasitic infections. It could help gauge what levels of systemic exposure occurs during Cryptosporidium therapy and compare that to toxicology measurements. Do you know anything about pharmacokinetics, in general, of TrtE. Or, if none of this is done/known, perhaps discussing this as something that should be done.

2) Can you tell the reader how reproducibly this natural product is produced by this bacterial strain and in what quantities? There are two descriptions of two batches; has more been made or attempted?

3) Can you tell the reader how difficult it appears to make sufficient quantities of TrtE reproducibly, so it could be used practically and commercially? Of if not, what steps are anticipated to make this happen?

4) Has any whole animal toxicity testing been done with TrtE and how much of a safety index is there compared with dosages needed for effective Cryptosporidium therapy. Or, if not done, perhaps discussing this as something that should be done.

5) Do you know anything about TrtE activity against Toxoplasma gondii bradyzoites?

Reviewer #3: For the compound tartrolon E (trtE), it may be necessary to balancedly discuss some chemical features of trtE and analogs, and their known bioactivities.

For rationale to explore a shipworm symbiotic bacterium as a source of new antiparasitic compounds, the introduction and some other parts of the manuscript was too extensive and unnecessary. Some statements were not well justified, such as the implication that shipworm symbiont needed to produce antibiotics to inhibit gregarine parasites, as gregarines usually do not cause too much harm to their hosts such as mollusks (at least having not been defined). It could be simply competing with other bacteria for local niche colonization.

The concept of Cryptosporidium as a gregarine was wrong. Cryptosporidium is not a gregarine, although they both are branched at the base of the Apicomplexa. Cryptosporidium and gregarines are also highly divergent in metabolism.

For broad-spectrum of activity: In principle, all ionophores (structurally, tartrolons are ion-carriers) share the same mechanism of action for their bioactivity, i.e., disrupting ion balances across cell membranes by insertion into bio-membranes. Because their primary targets are not specific proteins or other molecules (as they act on bio-membranes), they typically have narrow margins of safety (as demonstrated by several anti-coccidial ionophore drugs). The authors may want to discuss the significance of the findings more balancely by highlighting these and related points.

Figure 3: Please mark parasite names in individual figures for easy reading (e.g., in Y-axis or appropriate position).

PLOS authors have the option to publish the peer review history of their article (what does this mean?). If published, this will include your full peer review and any attached files.

Reviewer #1: No

Reviewer #2: Yes: Wesley C. Van Voorhis

Reviewer #3: No
---

## [Editor Report · Decision Letter 1]

28 Apr 2020

Dear Dr O'Connor,

Thank you very much for submitting your manuscript "A symbiotic bacterium of shipworms produces a compound with broad spectrum anti-apicomplexan activity." for consideration at PLOS Pathogens. As with all papers reviewed by the journal, your manuscript was reviewed by members of the editorial board and by several independent reviewers. The reviewers appreciated the attention to an important topic. Based on the reviews, we are likely to accept this manuscript for publication, providing that you modify the manuscript according to the review recommendations.

The authors revised the manuscript based on the requests and suggestions by reviewers and editor.

They have significantly cleared up their figures and figure legends, now provide extensive discussion on toxicity, pharmacokinetics and potential for synthetic scale up. The also reworded several claims to better reflect the experimental evidence.

Some minor concerns and corrections that should be easy for the authors to fix in the submission of a final version:

line 216: cut 'another' as it is unclear whether coccidian refers to Toxoplasma or Cryptosporidium here ( a view no longer held by most in the field), just say the coccidian Sarcocystis

line 255: Some reviewers dearly hated the idea to call Cryptosporidium a gregarine but the authors call it gastrointestinal gregarine parasite in this headline. I suggest to call it a gastrointestinal parasite. Yes they are closer to gragarine than e.g. Toxoplasma is, but overall the phylogeny around that question is still somewhat debated. The editor has little interest in that debate and it's really not important to this paper. Plasmodium is not called a gregarine here -- but still killed by the drug. The authors have an interesting evolutionary model and they should be able to discuss that. They did, so let's keep the rest focused on what is well established, and that is that C. parvum is a gastrointestinal parasite. (BTW many gregarines are gastrointestinal parasites as well, just not of mammals).

line 276: were/was

line 313 structural details

300-316: so this paragraph is really confusing in that it concludes that "tatralons are quite variable" and then "tartralons are very similar to each other". Please work this over to a clean and consistent claim.

Sincerely,

Boris Striepen

Guest Editor

PLOS Pathogens

Vern Carruthers

Section Editor

PLOS Pathogens

Kasturi Haldar

Editor-in-Chief

PLOS Pathogens

orcid.org/0000-0001-5065-158X

Michael Malim

Editor-in-Chief

PLOS Pathogens

orcid.org/0000-0002-7699-2064

The authors revised the manuscript based on the requests and suggestions by reviewers and editor.

They have significantly cleared up their figures and figure legends, now provide extensive discussion on toxicity, pharmacokinetics and potential for synthetic scale up. The also reworded several claims to better reflect the experimental evidence.

Some minor concerns and corrections that should be easy for the authors to fix in the submission of a final version:

line 216: cut 'another' as it is unclear whether coccidian refers to Toxoplasma or Cryptosporidium here ( a view no longer held by most in the field), just say the coccidian Sarcocystis

line 255: Some reviewers dearly hated the idea to call Cryptosporidium a gregarine but the authors call it gastrointestinal gregarine parasite in this headline. I suggest to call it a gastrointestinal parasite. Yes they are closer to gragarine than e.g. Toxoplasma is, but overall the phylogeny around that question is still somewhat debated. The editor has little interest in that debate and it's really not important to this paper. Plasmodium is not called a gregarine here -- but still killed by the drug. The authors have an interesting evolutionary model and they should be able to discuss that. They did, so let's keep the rest focused on what is well established, and that is that C. parvum is a gastrointestinal parasite. (BTW many gregarines are gastrointestinal parasites as well, just not of mammals).

line 276: were/was

line 313 structural details

300-316: so this paragraph is really confusing in that it concludes that "tatralons are quite variable" and then "tartralons are very similar to each other". Please work this over to a clean and consistent claim.
---

## [Editor Report · Decision Letter 2]

5 May 2020

Dear Dr O'Connor,

We are pleased to inform you that your manuscript 'A symbiotic bacterium of shipworms produces a compound with broad spectrum anti-apicomplexan activity.' has been provisionally accepted for publication in PLOS Pathogens.

Best regards,

Boris Striepen

Guest Editor

PLOS Pathogens

Vern Carruthers

Section Editor

PLOS Pathogens

Kasturi Haldar

Editor-in-Chief

PLOS Pathogens

orcid.org/0000-0001-5065-158X

Michael Malim

Editor-in-Chief

PLOS Pathogens

orcid.org/0000-0002-7699-2064
---

## [Editor Report · Acceptance letter]

18 May 2020

Dear Dr O'Connor,

We are delighted to inform you that your manuscript, "A symbiotic bacterium of shipworms produces a compound with broad spectrum anti-apicomplexan activity.," has been formally accepted for publication in PLOS Pathogens.

Best regards,

Kasturi Haldar

Editor-in-Chief

PLOS Pathogens

orcid.org/0000-0001-5065-158X

Michael Malim

Editor-in-Chief

PLOS Pathogens

orcid.org/0000-0002-7699-2064